# Child Labor in Family Tobacco Farms in Southern Brazil: Occupational Exposure and Related Health Problems

**DOI:** 10.3390/ijerph182212255

**Published:** 2021-11-22

**Authors:** Anaclaudia Gastal Fassa, Neice Muller Xavier Faria, Ana Laura Sica Cruzeiro Szortyka, Rodrigo Dalke Meucci, Nadia Spada Fiori, Maitê Peres de Carvalho

**Affiliations:** 1Department of Social Medicine, Faculty of Medicine, Federal University of Pelotas, Pelotas 96030-000, Brazil; neicef@yahoo.com.br (N.M.X.F.); nsfiori@yahoo.com.br (N.S.F.); maite_carvalho@yahoo.com.br (M.P.d.C.); 2Psychology Course, Faculty of Medicine, Federal University of Pelotas, Pelotas 96030-000, Brazil; alcruzeiro@gmail.com; 3Faculty of Medicine, Federal University of Rio Grande, Rio Grande 96203-900, Brazil; rodrigodalke@gmail.com

**Keywords:** child labor, agriculture, rural health, cotinine, tobacco, pesticides, occupational health, epidemiology, cross-sectional studies

## Abstract

Tobacco farming is considered Hazardous Child Labor in Brazil. This study examined the work of children and adolescents in tobacco farming, characterizing the level of urinary cotinine and the occurrence of Green Tobacco Sickness (GTS), pesticide poisoning, respiratory symptoms, and musculoskeletal disorders. A cross-sectional descriptive study was conducted with a random sample of tobacco growers under 18 years old in Southern Brazil. Ninety-nine young people were interviewed at 79 family farms. The majority began working in agriculture before they were 14 and worked harvesting and tying hands of tobacco; 60% were 16 or 17 years old, and 51.5% were male. During their lifetime, 24.5% reported GTS, and 3% reported pesticide poisoning. In the previous year, 29.3% reported low back pain, 6.1% wheezing, and 16.2% coughing without having a cold. Half of the 12 young people evaluated had over 100 ng/mL of urinary cotinine. The study indicates that child laborers do various activities and present a high prevalence of health problems. Health workers should be trained to identify child laborers and their impacts on health. Full-time farm schools could provide knowledge about sustainable agricultural production, reducing the rates of age-grade mismatch, without taking young people away from rural areas.

## 1. Introduction

Brazil is the world’s second largest producer and the largest exporter of tobacco. In the 2019–2020 harvest, the southern region of the country produced more than 600,000 tons of tobacco, involving approximately 146,000 families [1]. Child labor in agriculture and tobacco growing is characterized as hazardous according to Brazilian law [2,3], constituting a violation of human rights, perpetuating the cycle of poverty, contributing to lost educational opportunities, and gender inequality [4,5,6].

According to the Instituto Brasileiro de Geografia e Estatística (IBGE), in 2019, the occurrence of child labor in Brazil was 4.6% among children and adolescents aged 5 to 17 years (1.8 million). Of these workers, 53.7% were 16 and 17 years old, and 66.4% were male. Among the youngest, aged 5 to 13, 16.9% worked more than 14 h per week, while those between 16 and 17 years old, 24.2% worked more than 40 h per week [7]. In 2016, 47.6% of workers aged 5 to 13 worked in agriculture [8].

Studies with adults indicate that work in agriculture, especially during the harvest period, requires long working days, working in awkward positions, with repetitive movements and physical exertion [9,10,11,12,13], involving exposure to pesticides, dust, and weather conditions such as heat, hail, frost, and storms [14,15,16]. In addition to the above, tobacco growing also includes exposure to nicotine [17,18]. These forms of exposure have been associated with mental health, respiratory and musculoskeletal problems, pesticide poisoning, and green tobacco sickness (GTS) [11,12,13,14,16,19,20,21].

In the context of child labor, a study conducted in Nicaragua found that 21% of acute pesticide poisoning cases are related to occupational activities, and the work circumstances related to poisonings were similar to those found in adults, such as picking freshly sprayed tobacco leaves, using spraying equipment in bad condition with backpack leakage, preparing pesticide solutions and not using personal protective equipment [22]. In addition, studies on tobacco farming work indicate that age is inversely associated with GTS, since with the prolonged exposure, the workers develop nicotine tolerance [18,19]. 

There are only a few studies evaluating the respiratory health of children working in agriculture; among them, a study in North Carolina found that 56.4% of the children had at least one respiratory problem [21]. Within the overall scope of the present study, analyses of adult workers indicate that 21.2% reported chest pain [12] and 7.4% reported cervical spine pain [11]. Among adults with chronic low back pain, 37.6% were unable to carry out some jobs [23]. Doing tasks in awkward positions and carrying heavy loads typical of tobacco growing have been associated with an increase in musculoskeletal disorders [10,11,12]. Quandt et al. (2021) [13] indicate that 21.3% of boys working in agriculture had significantly more knee injuries than girls (4.1%). Children and adolescents may be more vulnerable to some forms of exposure than adults [24,25].

Considering the prevalence of child labor in agriculture, its classification as hazardous child labor, and the scarcity of studies on the subject in Brazil and worldwide, mainly focused on tobacco farming, this study examined the work of children and adolescents in tobacco farming, characterizing the level of urinary cotinine and the occurrence of GTS, pesticide poisoning, respiratory symptoms, and musculoskeletal disorders.

## 2. Materials and Methods

A cross-sectional study was conducted with a random sample of tobacco farmers in the municipality of São Lourenço do Sul, southern Brazil, during the 2011 harvest (January to March). Within this project, 2469 workers were studied; however, only individuals under 18 years old were included in the present article. The southern Brazilian region is responsible for over 95% of the country’s tobacco production; the great majority of the population is of German origin, and its economy is based on tourism and family farming, primarily focused on Virginia tobacco growing [1].

The sample was selected from 3851 invoices provided by the Municipal Treasury Department, referring to tobacco sales and issued in 2009. We estimated about three workers per farm and obtained a simple random sample of 1100 invoices. Tobacco farmers who worked at least fifteen hours per week were eligible for the study. If a selected property was no longer producing tobacco, it was replaced by the nearest tobacco property. Workers residing in the urban area of the municipality or who no longer lived in the municipality were considered ineligible. The selected properties were identified with the help of Community Health Agents from the rural area of São Lourenço do Sul. At the end of the study, 912 properties were identified, and 99 young people from 79 of these properties were interviewed.

Regarding data collection, interviewers specifically trained for the study used PDAs (personal digital assistants) to apply two electronic instruments. One instrument assessed aspects of the farm and was answered by the farm manager, and the other focused on individual aspects, collecting data on sociodemographic and behavioral issues, work activities, workloads, GTS, pesticide poisoning, respiratory symptoms, and musculoskeletal pain. In addition, urinary cotinine samples were collected. Selected questions used to assess health problems can be viewed in the Appendix A.

The schooling variable was categorized according to age–grade mismatch parameters established by the United Nations Children’s Fund (UNICEF). The mismatch was calculated in years and represented the gap between the student’s age and the recommended age for the grade he or she is attending. When the difference between the student’s age and expected age for the grade was two years or more, this was considered an “age-grade mismatch” [26]. Workers who smoked one or more cigarettes per day of any type during the last month were considered smokers, while those who reported having quit smoking more than one month ago were considered former smokers. Passive smokers were those who had people close to them who frequently smoked in their presence. In order to characterize alcoholic beverage intake, we asked about consumption on weekdays (Monday through Friday) and at the weekend (Saturdays and Sundays) in the past 30 days, considering a standard dose to be equivalent to about 10–12 g of pure ethanol, as established by the World Health Organization [27].

The presence of musculoskeletal pain was assessed by the Nordic Musculoskeletal Questionnaire [28], which has been validated in Brazil [29] and has already been used in other studies [30,31]. Each participant was asked if they had had back pain in the 12 months prior to the interview. To those who reported pain, a picture of the human body standing in an upright position was shown, with the cervical, thoracic, and lumbar regions highlighted to identify the site(s) of pain. Pesticide poisoning was self-reported by the workers. GTS during lifetime was defined as the occurrence of dizziness or headache and nausea or vomiting up to two days after handling tobacco [32].

Urinary cotinine was measured in workers with or without symptoms of GTS, who worked on farms located in the two districts with the highest tobacco production in São Lourenço do Sul, and who had applied pesticides in the year before the survey. Cotinine is the primary metabolite of nicotine and the most suitable parameter to assess exposure to tobacco, as it has greater stability and a longer half-life than nicotine [33,34]. Urine samples were stored in a freezer at −10 °C for conservation and sent weekly to the Municipal Health Department of São Lourenço do Sul, which would transport the samples to the Toxicology Laboratory of the Pontifical Catholic University of Rio Grande do Sul in the city of Porto Alegre/RS. The High-Performance Liquid Chromatography (HPLC) method with an ultraviolet detector was used, and the urinary cotinine concentration was calculated by its absorbance at 260 nm, considering the peak area and the calibration curve (r > 0.99), using an already standardized technique [35]. With the urinary cotinine among nonsmokers as a reference (lower than 20 ng/mL), the concentration was categorized as less than 20 ng/mL, 20–100 ng/mL, and more than 100 ng/mL.

Data analysis was performed using Stata 12.0^®^ (StataCorp LLC, College Station, TX, USA). Descriptions of continuous variables were evaluated through their means. The frequency of the categorical variables was verified by the prevalence and confidence intervals (95%CI), stratified by sex and age. In order to assess differences by age group and sex for each variable, the Chi-square Test for Heterogeneity was used, and, in the case of small numbers (cells with less than five observations), the Fisher’s Exact Test was utilized. Missing data were excluded from the analysis.

This study was approved by the Research Ethics Committee of the Federal University of Pelotas (Protocol Code no. 11/2010). The participants’ legal guardians and the young workers themselves were informed about the research topic, the right not to participate or to withdraw from participating at any time, and the confidentiality of individual information. Those who agreed to participate signed the Informed Consent Form.

## 3. Results

The sample was comprised of 99 workers of both sexes, under 18 years old, who worked in 79 family properties. These farms had on average 37.3 hectares (95%CI 24.0–50.6), of which 7.5 hectares (95%CI 5.9–9.1) were used for tobacco growing. The majority of the farm properties (72.7%) had at least five farming machines, and 83.2% had a tractor. About 93% of the farmers worked on farms with at least one vehicle (motorcycle, car, pickup truck, or truck). The main pesticides used on these farms were: herbicides (clomazone 72.6%, glyphosate 65.3%, sulfentrazone 48.4%), insecticides (neonicotinoids 76.8%, organophosphates 63.2%, pyrethroids 38.9%), fungicides (dithiocarbamates 65.3%, metalaxyl 49.5%, iprodione 33.7%), and growth regulators (flumetralin 92.6%).

Among the 99 young people, 60% were between 16 and 17 years old, 51.5% were male, around 10% were smokers or former smokers, and almost 60% reported being passive smokers. As for alcohol consumption at the weekend, more than 60% of the 16 and 17 years old, and almost 25% of the 14 and 15 years old, reported drinking one or more doses daily. During the week, around 13% of the young respondents reported consuming alcoholic beverages occasionally. The age–school grade mismatch was 3.5% among 14 and 15 years old, and 44.1% among 16 and 17 years old. Approximately 80% of the young people worked on farm properties that produced up to 10,000 kg of tobacco, with those under 16 years of age concentrated in small and medium-sized farm properties. Almost all respondents had some type of curing barn on their farm property (98%), and 28.4% had an electric curing barn. Passive smoking (*p* = 0.037) and alcohol consumption at the weekend (*p* = 0.005) were higher among male workers. Among the young male workers, 76% worked at farms producing less than 10,000 kg of tobacco, and 88.8% of females also worked at farms of this size (*p* = 0.043) (Table 1).

Among the young people interviewed, 66% began doing farm work before they were 14 years old, approximately 14% began working with pesticides at 14–15 years old and 52.5% dedicated at least 5 h per day to farm work outside the harvest period. Among those under 14 and those who were 14 and 15 years old, 36.4% and 41.3%, respectively, worked more than 4 h a day doing farming activities outside the harvest period. Among those who were 16 and 17 years old, 39% worked more than 7 h a day. During the harvest, 84.6% of the young people worked more than 4 h a day, and among those older than 13, 65.9% worked more than 7 h a day. Approximately 20% of the young people did not perform domestic work, while 34.3% and 20.2%, respectively, spent more than 3 h a day doing domestic work both outside and during the harvest. More than 70% of those under 14 years old had less than 11 h of leisure time both during and outside the harvest, while among those who were 14 to 17 years old, 59% and 48.8%, respectively, had more than 10 h of leisure time per week, outside the harvest and during the harvest (Table 2).

When stratifying by sex, we found that boys worked more with pesticides (47.1%) than girls (12.5%). Domestic work for more than 3 h a day was more frequently carried out by girls, both outside (54.1%) and during harvest time (33.3%) (Table 2).

The activities undertaken varied little according to age group. Most young people were involved in tying hands of tobacco (95%), harvesting tobacco in the last week (85.7%), tending animals (83.8%), transplanting tobacco (82.2%), lifting tobacco sticks (78.8%), and sowing tobacco (74.7%). About 60% were also involved in carrying and transporting green leaves, baling tobacco, harvesting wet leaves, and grading tobacco leaves. More than half of the young people drove tractors or other farm machinery, 37.4% climbed high into the curing barn, and 23.2% drove cars or trucks. These activities were mainly performed among 16 and 17 years old and less frequently by those younger than 16. Controlling the temperature of curing barns during the night was performed by about 10% of the young people. Among the 14 to 17 years old, 17% cut trees with a chainsaw, and 15.9% reported carrying more than 50 kg. Among the 16 and 17 years old, 13.6% carried bales weighing about 50 kg. More than 80% of the 14 to 17 years old, and 27.3% of those under 14 years old, worked in a bending position. Wearing protective clothing during tobacco harvesting was reported by 38.4% of the individuals interviewed, and wearing gloves for harvesting was mentioned by 49.5% (Table 3).

When stratifying by sex, the following activities were performed more by boys: sowing tobacco (88.2%), carrying and transporting green leaves (54.9%), climbing high into the curing barn (54.9%), driving tractors/farm machinery (88%), driving cars/trucks (41.2%), cutting trees with chainsaw (30.6%), delimbing trees (47.1%), controlling curing barn temperature during the day (47.1%) and at night (17.6%) and carrying weight (55%). Looking after the vegetable garden (54.2%) was more frequent among girls (Table 3).

Regarding tobacco plantation activities involving exposure to pesticides, we found that children under 14 did not engage in this type of activity. Among 16 and 17 years old, approximately 18% prepared pesticide solution and applied pesticides, 12.1% went into the plantation shortly after pesticide application, and 8.6% cleaned pesticide application equipment. The main activities involving pesticide exposure carried out by 14 and 15 years old included washing clothes worn during pesticide application (13.8%) and re-entering the treated field shortly after pesticide application (6.9%). Preparing pesticide mixture (*p* = 0.013), applying pesticides (*p* = 0.043), and cleaning equipment (*p* = 0.040) were performed more frequently by boys while washing clothes worn during pesticide application was more frequent among girls (*p* = 0.010) (Table 4).

Urinary cotinine was analyzed in 12 of the 99 young people, all of them nonsmokers; 33.3% had 20 to 100 ng/mL, and 50% had more than 100 ng/mL of cotinine in their urine (Table 5).

Approximately 25% of the interviewees reported an occurrence of GTS once during their lifetime, and 13.3% reported having had it three times or more. Among workers aged 14 to 17, 3.5%, all of whom were male, reported pesticide poisoning during their lifetime (Table 5).

Regarding respiratory symptoms, 16.2% of the young people reported coughing without having a cold, and 6.1% reported wheezing in the last 12 months. As for musculoskeletal disorders, low back pain was the most frequent, affecting 29.3% of the workers and increasing with age, with 39% frequency among workers aged 16 and 17. Thoracic spine pain in the last year was reported by 26.3% of the workers. Neck pain in the last year was reported by 3.0%, all of whom were female (Table 5). Figure 1 shows the prevalence of health problems.

## 4. Discussion

Child labor in tobacco farming is prohibited for anyone under 18 years old [3]. Notwithstanding, in 79 of the 912 properties studied, mostly small and medium-sized family farms, there were young people under 18 years old who had worked in the year prior to the interview. The majority started working when they were less than 15 years old, for more than 5 h a day, increasing the time they spent working during the harvest period. There was also a high prevalence of age–grade mismatches among 16 and 17 years old. The activities which these young people carried out most, tying hands of tobacco and picking, occurred during the harvest, which coincides with school holidays. In addition, the young people had performed a variety of activities both during the harvest and during the rest of the year, including some high-risk activities such as working with pesticides, driving tractors, and cutting trees with chainsaws.

A study conducted by Arcury et al. (2016) [36] indicated that tobacco farmers had much higher levels of urinary cotinine (396.03 ng/mL) when compared to individuals who did not work in tobacco farming (9.03 ng/mL). Exposure to green tobacco leaves causes transdermal absorption of nicotine, thus increasing nicotine levels in the blood and resulting in GTS. Nicotine is water-soluble [36], and as such, its absorption increases when harvesting leaves are wetted by morning dew or due to contact with transpiration. Absorption is also increased by not using personal protective equipment, by eventual impairment of the skin integrity, and vasodilation resulting from working in hot and humid environments [37,38,39,40,41].

In the present study, the strong participation of young people in tying hands of tobacco and harvesting, with significant exposure to nicotine, is reflected in the percentage of those who had detectable urinary cotinine and had already had GTS. There are indications that workers who are more exposed and who are smokers can develop nicotine tolerance. Thus, younger workers [19], children and adolescents [42], individuals with less working time [32,42], and nonsmokers [19,32,42] are at greater risk of developing GTS. The consequences of massive nicotine exposure in the medium and long term are still little studied. Within the same study, when analyzing tobacco growers 18 years or older, a positive linear association of the number of GTS episodes with mental disorders and with respiratory symptoms was confirmed [14,20].

Studies with farmers indicate that exposure to pesticides has both acute effects, such as poisoning, as well as long-term effects. The effects of pesticides on health vary according to the type of chemical used, and multi-chemical exposure makes them more complex to assess. The most studied pesticides are organophosphates, which are neurotoxic and have been associated with mental health problems such as depression [14,16,43,44,45] and cognitive deficits [46]. Other analyses within the overall scope of this study indicated that pesticide poisoning doubled the risk of suicidal ideation [47]. Children and adolescents are probably more susceptible to chemical exposure since they are in a period of maturation of their endocrine system, and such exposure could alter the delicate hormonal balance and their biofeedback mechanisms [24,25]. In the present study, children over 13 years old carried out several activities involving pesticide exposure and reported episodes of acute pesticide poisoning. Thus, the potential long-term health effects of pesticides on children and adolescents are of concern.

Tying hands of tobacco is one of the main activities undertaken by young people in tobacco farming. In general, it is done inside the barn so that workers are exposed to organic dust that increases the risk of respiratory problems [48]. A North American study [21] with children who worked in agriculture found that 16.4% had wheezing at some time in their lives, 14.3% had a persistent cough for more than ten days, and one-third had two or more respiratory problems. The prevalence of coughing without having a cold in the present study was higher than in the North American study [21] and, considering the difference in recall, 6.1% prevalence of wheezing in the last year was considerable. Association between respiratory problems and contact with dust, pesticides, and microorganisms is well established [15,49,50,51,52]. There is evidence that inhaling nicotine is associated with respiratory problems [53,54]; thus, the nicotine in tobacco leaf dust may increase the risk of this type of problem. However, there are no studies that assess whether the high dermal absorption of nicotine also potentiates the risk of respiratory problems [20].

Reis et al. (2017) [10] emphasize that tobacco growing in family farming requires manual work and often exposes workers to repetitive movements, excessive weightlifting, and inappropriate postures. Given the activities performed, work-related musculoskeletal disorders in the shoulder, elbow, knee, and hip, as well as in the different segments of the spine, become frequent and are evidenced by several studies with farmers of different ages [9,11,12,13]. Harvesting and tying hands of tobacco involves repetitive movements and bending and sitting postures. Other activities such as carrying weight (predominantly done by boys) and lifting tobacco sticks also place heavy demands on the musculoskeletal system because apart from the posture and movement of the upper limbs, they demand physical strength.

Quandt et al. (2021) [13] found that 6.4% of young people performing farm work reported low back pain in at least one site during physical examination. In the present study, more than one-third of young people aged 16–17 reported low back pain in the last year. This may be related to greater involvement with work and the fact that they have been working longer when compared with younger children. There is a concern that children and adolescents are more vulnerable than adults to ergonomic exposures since, as they are in a period of rapid growth, they could be at increased risk of injuries to bone epiphyses or ligaments. In addition, the long-term effect of ergonomic exposure on musculoskeletal disorders is unclear [24,25].

It is important to consider that these aspects are aggravated by long working hours typical of the harvest period. Quandt et al. (2021) [13] indicate that children involved in agricultural work perform less intense tasks and for less time than adults, corroborating what was found in the present study, namely that children and adolescents worked for fewer hours outside the harvest period and that their involvement in different farming tasks increased with age. However, during the harvest, most adolescents work in a similar way to adults, performing diversified tasks and working more than seven hours a day.

Among the limitations of the present study, it is important to recognize the occurrence of selection bias due to the fact that working in the tobacco industry is prohibited for people under 18 years old [3]. This certainly caused child labor to be partly hidden, preventing the identification of young workers by the study. The illegal nature of this work may also have caused information bias, underestimating work in activities of higher risk such as exposure to pesticides and driving a tractor, among others. On the other hand, it is possible that the young people identified are those who participate more strongly in production activities. The urinary cotinine samples were obtained just from 12 young workers, and pesticide poisoning was not evaluated through biomarkers but rather characterized by self-reported information. These aspects limited the chemical exposure and related outcomes evaluation.

Data collection for this study took place in 2011. Since then, there have been no considerable changes in the production system that could substantially alter tobacco farming. However, there were contextual variations. Considering the reduction in tobacco consumption and the production migration to low-income countries increasing competition, policies were implemented to encourage crop diversification. In addition, the COVID-19 pandemic has caused schools to suspend in-person activities and may have led tobacco producers to avoid hiring temporary workers. These aspects can modify both the prevalence of child labor and the type of activity that children carry out in tobacco production.

## 5. Conclusions

Despite the prohibition, this study pointed out the persistence of young people under 18 years old working in tobacco growing, in various activities, the most frequent being harvesting tobacco leaves and tying hands of tobacco. Chemical exposure to nicotine and pesticides, ergonomic exposure, such as working in awkward positions, and physical exposure, such as exposure to dust, resulted in a high prevalence of GTS, pesticide poisoning, respiratory, and musculoskeletal problems.

Future studies on child and adolescent populations in relation to tobacco growing are needed to assess the health of workers and nonworkers with larger sample sizes. Nonworkers may also be exposed to some extent by being with their parents on the plantation or in the barn. In addition, studies should intensify objective assessment of exposure to dust, nicotine, and pesticide, as well as assessment of outcomes such as pesticide poisoning and lung function.

Considering that working in tobacco growing takes place in the context of family farming, it is fundamental to provide clarification to both adults and young people about the forms of exposure present in this type of work and its effect on health. Health workers are privileged actors and should be trained to identify and report situations of child labor in tobacco farming. They should also be prepared to recognize GTS, pesticide poisoning, as well as respiratory and musculoskeletal problems and relate them to this form of work, providing relevant guidance for rehabilitation and prevention of new episodes. The implementation of full-time agricultural schools can provide knowledge about sustainable agricultural production models, reducing the rates of age–grade mismatch without taking young people away from rural areas.

## Figures and Tables

**Figure 1 ijerph-18-12255-f001:**
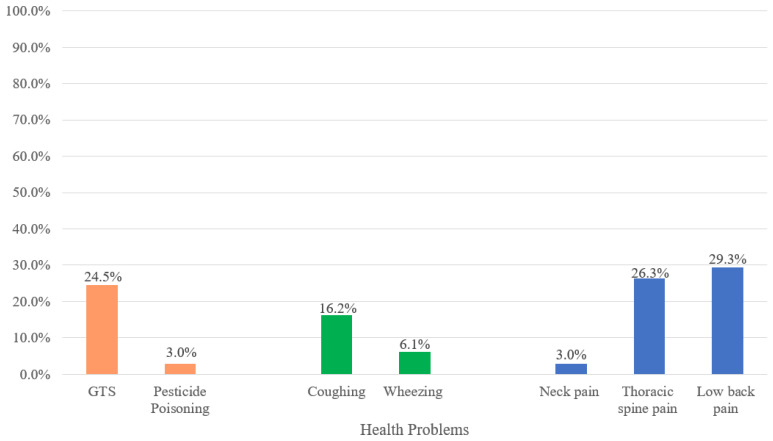
Prevalence of Health Problems among Children Tobacco Growers. São Lourenço do Sul, Rio Grande do Sul, Brazil, 2011 (*n* = 99).

**Table 1 ijerph-18-12255-t001:** Demographic, socioeconomic, and behavioral characterization of tobacco farmers stratified by age and sex. São Lourenço do Sul, Rio Grande do Sul, Brazil, 2011. (*n* = 99).

Variables	Total (*n* = 99)	Age Group				Sex		
		<14 Years (*n* = 11)	14–15 Years (*n* = 29)	16–17 Years (*n* = 59)	*p*-Value for Age	Male (*n* = 51)	Female (*n* = 48)	*p*-Value for Sex
	%	%	%	%		%	%	
Sex					0.975 ^a^			---
Male	51.5	54.5	51.7	50.8		---	---	
Female	48.5	45.5	48.3	49.2		---	---	
**Age-school grade mismatch ***					<0.001 ^b^			0.065 ^a^
Adequate for age	72.7	100	96.5	55.9		64.7	81.3	
Inadequate for age	27.3	0	3.5	44.1		35.3	18.7	
**Smoking**					0.252 ^b^			0.117 ^b^
No	92.9	90.9	100	89.8		88.2	97.9	
Yes	4.1	0	0	6.8		7.8	0	
Former smoker	3.0	9.1	0	3.4		4.0	2.1	
**Passive smoking**					0.913 ^b^			0.037 ^b^
No	43.4	36.4	44.8	44.1		33.3	54.2	
Yes	56.6	63.6	55.2	55.9		66.7	45.8	
**Alcohol consumption at weekends**					0.007 ^b^			0.005 ^b^
Has never drunk	33.3	54.5	44.9	23.7		25.5	41.6	
Drinks occasionally	20.2	27.3	31.0	13.6		21.6	18.8	
≤1 dose/day	16.1	0	13.8	20.4		13.7	18.8	
2 doses/day	15.2	9.1	0	23.7		11.8	18.8	
3 or more doses/day	15.2	9.1	10.3	18.6		27.4	2.0	
**Alcohol consumption during the week**					0.977 ^b^			0.202^b^
Has never drunk	82.8	90.9	79.2	83.0		76.5	89.6	
Drinks occasionally	13.1	9.1	13.8	13.6		15.7	10.4	
≤1 dose/day	2.0	0	3.5	1.7		3.9	0	
2 or more doses/day	2.0	0	3.5	1.7		3.9	0	
**Amount of tobacco produced (Kg) ****					0.074 ^b^			0.043 ^b^
1–5000	32.6	20.0	37.0	32.8		22.0	44.4	
5001–10,000	49.5	80.0	55.6	41.3		54.0	44.4	
10,001–36,000	17.9	0	7.4	25.9		24.0	11.2	
**Curing barn on the property ****					0.756 ^b^			0.433 ^b^
No curing barn	2.1	0	3.7	1.7		2.0	2.2	
Only conventional curing barn	69.5	70.0	74.1	67.3		64.0	75.6	
Only electric curing barn	10.5	0	7.4	13.8		10.0	11.1	
Both	17.9	30.0	14.8	17.2		24.0	11.1	

* Age–school grade mismatch in Brazil established by the United Nations Children’s Fund (UNICEF). ** 4 missing data for these variables. ^a^ Chi-square Test for Heterogeneity. ^b^ Fisher’s Exact Test.

**Table 2 ijerph-18-12255-t002:** Organization and division of tobacco farmers’ work, stratified by age and sex. São Lourenço do Sul, Rio Grande do Sul, Brazil, 2011. (*n* = 99).

Variables	Total (*n* = 99)	Age Group				Sex		
		<14 Years (*n* = 11)	14–15 Years (*n* = 29)	16–17 Years (*n* = 59)	*p*-Value for Age	Male (*n* = 51)	Female (*n* = 48)	*p*-Value for Sex
	%	%	%	%		%	%	
**Age at which began doing farm work ***					0.049 ^b^			0.752 ^b^
<10 years	3.3	0	3.6	3.5		4.2	2.3	
10–11 years	23.1	83.3	14.3	21.1		22.9	23.3	
12–13 years	39.5	16.7	53.5	35.1		43.7	34.9	
14–15 years	33.0	0	28.6	38.5		29.2	37.2	
16–17 years	1.1	0	0	1.8		0	2.3	
**Age at which began working with pesticide**					0.132 ^b^			0.001 ^b^
Does not work with pesticide	69.7	81.8	72.4	66.0		52.9	87.5	
<14 years	9.1	18.2	13.8	5.1		11.8	6.2	
14–15 years	14.1	0	13.8	17.0		23.5	4.2	
16–17 years	7.1	0	0	11.9		11.8	2.1	
**Daily hours of farm work outside harvest period**					0.095 ^b^			0.061 ^b^
Does not work	10.1	0	10.3	11.9		5.9	14.6	
<5 h	37.4	63.6	48.4	27.1		29.4	45.8	
5–7 h	23.2	27.3	24.1	22.0		25.5	20.8	
>7 h	29.3	9.1	17.2	39.0		39.2	18.8	
**Daily hours of farm work during the harvest**					0.006 ^b^			0.681 ^a^
<5 h	15.2	9.1	27.6	10.2		11.8	18.8	
5–7 h	23.2	63.6	13.8	20.3		23.5	22.9	
>7 h	61.6	27.3	58.6	69.5		64.7	58.3	
**Daily hours of domestic work outside harvest period**					0.839 ^b^			<0.001 ^b^
Does not work	24.2	27.3	27.6	22.0		41.2	6.3	
1–3 h	41.4	27.3	41.4	44.1		43.1	39.6	
>3 h	34.3	45.4	31.0	33.9		15.7	54.1	
**Daily hours of domestic work during the harvest**					0.908 ^b^			<0.001 ^b^
Does not work	22.2	18.2	27.6	20.3		41.2	2.1	
1–3 h	57.6	54.5	55.2	59.4		21.0	64.6	
>3 h	20.2	27.3	17.2	20.3		7.8	33.3	
**Weekly hours of leisure outside harvest period**					0.229 ^b^			0.619 ^b^
<6 h	11.1	27.3	6.9	10.2		13.7	8.3	
6–10 h	33.3	45.4	31.0	32.2		29.4	37.5	
>10 h	55.6	27.3	62.1	57.6		56.9	54.2	
**Weekly hours of leisure during the harvest**					0.343 ^b^			0.523 ^a^
<6 h	15.2	18.2	17.2	13.6		13.7	16.7	
6–10 h	39.4	63.6	34.5	37.3		35.3	43.7	
>10 h	45.4	18.2	48.3	49.1		51.0	39.6	

* 8 missing data for this variable. ^a^ Chi-square Test for Heterogeneity. ^b^ Fisher’s Exact Test.

**Table 3 ijerph-18-12255-t003:** Tobacco worker activities and workloads, stratified by age group and sex. São Lourenço do Sul, Rio Grande do Sul, Brazil, 2011. (*n* = 99).

Variables	Total (*n* = 99)	Age Group				Sex		
		<14 Years (*n* = 11)	14–15 Years (*n* = 29)	16–17 Years (*n* = 59)	*p*-Value for Age	Male (*n* = 51)	Female (*n* = 48)	*p*-Value for Sex
	%	%	%	%		%	%	
**Sowing tobacco**					0.436 ^b^			0.001 ^a^
No	25.3	18.2	34.5	22.0		11.8	39.6	
Yes	74.7	81.8	65.5	78.0		88.2	60.4	
**Transplanting tobacco**					0.063 ^b^			0.686 ^a^
No	17.2	27.3	27.6	10.2		15.7	18.7	
Yes	82.2	72.7	72.4	89.8		84.3	81.3	
**Harvesting tobacco in the last week ***					0.913 ^b^			0.934 ^a^
No	14.3	18.2	14.3	13.6		14.0	14.6	
Yes	85.7	81.8	85.7	86.4		86.0	85.4	
**Harvesting wet leaves**					0.537 ^a^			0.287 ^a^
No	38.4	45.5	44.8	33.9		33.3	43.7	
Yes	61.6	54.5	55.2	66.1		66.7	56.3	
**Carrying and transporting green leaves**					0.227 ^b^			<0.001 ^a^
No	37.4	45.5	48.3	30.5		45.1	79.2	
Yes	62.6	54.6	51.7	69.5		54.9	20.8	
**Lifting tobacco sticks** (Around 12–14 Kg)					0.292 ^b^			0.127 ^a^
No	21.2	36.4	13.8	22.0		54.9	39.6	
Yes	78.8	63.6	86.2	78.0		45.1	60.4	
**Grading tobacco leaves**					0.915 ^a^			0.510 ^a^
No	40.4	45.5	41.4	39.0		37.3	43.8	
Yes	59.6	54.5	58.6	61.0		62.7	56.2	
**Tying hands of tobacco**					1.000 ^b^			0.363 ^b^
No	5.0	0	3.5	6.8		7.8	2.1	
Yes	95.0	100	96.5	93.2		92.2	97.9	
**Baling tobacco**					0.196 ^a^			<0.001 ^a^
No	37.4	54.5	44.8	30.5		19.6	56.3	
Yes	62.6	45.5	55.2	69.5		80.4	43.7	
**Transporting tobacco bales** (around 50 Kg)					0.065 ^b^			0.270 ^b^
No	91.9	100	100	86.4		88.2	95.8	
Yes	8.1	0	0	13.6		11.8	4.2	
**Climbing high into the curing barn**					0.231 ^b^			<0.001 ^a^
No	62.6	81.8	51.7	64.4		45.1	81.3	
Yes	37.4	18.2	48.3	35.6		54.9	18.7	
**Taking care of the vegetable garden**					0.051 ^a^			0.007 ^a^
No	59.6	45.5	44.8	69.5		72.6	45.8	
Yes	40.4	54.5	55.2	30.5		17.4	54.2	
**Tending animals**					0.004 ^b^			0.076 ^b^
No	16.2	18.2	34.5	6.8		9.8	22.9	
Yes	83.8	81.8	65.5	93.2		90.2	77.1	
**Driving tractor/farm machines ***					0.443 ^b^			<0.001 ^a^
No	46.9	63.6	48.3	43.1		22.0	72.9	
Yes	53.1	36.4	51.7	56.9		88.0	27.1	
**Driving car/truck**					0.645 ^b^			<0.001 ^b^
No	76.8	81.8	82.8	72.9		58.8	95.8	
Yes	23.2	18.2	17.2	27.1		41.2	4.2	
**Cutting trees ***					0.293 ^b^			<0.001 ^b^
Does not cut	78.1	100	77.8	74.1		61.2	95.7	
Cut with chainsaw	15.6	0	11.1	20.7		30.6	0	
Cut with other equipment	6.3	0	11.1	5.2		8.2	4.3	
**Delimbing trees**					0.046 ^b^			<0.001 ^a^
No	68.7	100	65.5	64.4		52.9	85.4	
Yes	31.3	0	34.5	35.6		47.1	14.6	
**Controlling the temperature of the curing barn during the day**					0.077 ^a^			0.023 ^a^
No	63.6	45.5	79.3	59.3		52.9	75.0	
Yes	36.4	54.5	20.7	40.7		47.1	25.0	
**Controlling the temperature of the curing barn during the night**					0.206 ^b^			0.016 ^b^
No	89.9	81.8	96.5	88.1		82.4	97.9	
Yes	10.1	18.2	3.5	11.9		17.6	2.1	
**Maximum weight usually carried**					0.393 ^b^			<0.001 ^b^
Does not carry weight	61.7	81.8	69.0	54.2		45.0	79.2	
Up to 49 Kg	24.2	18.2	17.2	28.8		27.5	20.8	
50 Kg or over	14.1	0	13.8	17.0		27.5	0	
**Working in a bending position**					<0.001 ^b^			0.071 ^a^
No	21.2	72.7	13.8	15.3		17.6	25.0	
Yes	78.8	27.3	86.2	84.7		82.4	75.0	
**Wearing protective clothing during harvest**					0.102 ^b^			0.812 ^a^
No	61.6	90.9	55.2	59.3		62.8	60.4	
Yes	38.4	9.1	44.8	40.7		37.2	39.6	
**Wearing gloves during harvest**					0.699 ^b^			0.088 ^a^
No	50.5	63.6	48.3	49.1		58.8	41.7	
Yes	49.5	36.4	51.7	50.9		41.2	58.3	

* 3 missing data for these variables. ^a^ Chi-square Test for Heterogeneity. ^b^ Fisher’s Exact Test.

**Table 4 ijerph-18-12255-t004:** Activities involving exposure to pesticides among tobacco farmers who worked with pesticides in the last 12 months. São Lourenço do Sul, Rio Grande do Sul, Brazil, 2011. (*n* = 99).

Variables	Total (*n* = 99)	Age Group				Sex		
		<14 Years (*n* = 11)	14–15 Years (*n* = 29)	16–17 Years (*n* = 59)	*p*-Value for Age	Male (*n* = 51)	Female (*n* = 48)	*p*-Value for Sex
	%	%	%	%		%	%	
**Preparing mixture ***					0.080 ^b^			0.013 ^b^
Does not work with pesticide	50.0	81.8	48.3	44.8		42.0	58.3	
Does not prepare	38.8	18.2	48.3	38.0		38.0	39.6	
Prepares	11.2	0	3.4	17.2		20.0	2.1	
**Applying ***					0.059 ^b^			0.043 ^b^
Does not work with pesticide	50.0	81.8	48.3	44.8		42.0	58.3	
Does not apply	37.8	18.2	48.3	36.2		38.0	37.5	
Applies	12.2	0	3.4	19.0		20.0	4.2	
**Cleaning equipment used to apply ***					0.112 ^b^			0.040 ^b^
Does not work with pesticide	50.0	81.8	48.3	44.8		42.0	58.3	
Does not clean	44.9	18.2	51.7	46.6		48.0	41.7	
Cleans	5.1	0	0	8.6		10.0	0	
**Washing clothes contaminated**					0.123 ^b^			0.010 ^b^
Does not work with pesticide	50.0	81.8	48.3	44.8		42.0	58.3	
Does not wash	42.9	18.2	37.9	50.0		56.0	29.2	
Washes	7.1	0	13.8	5.2		2.0	12.5	
**Re-entering the treated field ***					0.281 ^b^			0.261 ^b^
Does not work with pesticide	50.0	81.8	48.3	44.8		42.0	58.3	
Does not go in	40.8	18.2	44.8	43.1		46.0	35.4	
Goes in	9.2	0	6.9	12.1		12.0	6.3	

* 1 missing data for these variables. ^b^ Fisher’s Exact Test.

**Table 5 ijerph-18-12255-t005:** Health outcomes among tobacco farmers stratified by age group and sex. São Lourenço do Sul, Rio Grande do Sul, Brazil, 2011.

**Variables**	**Total** **(*n* = 12)**	**Age Group**				**Sex**		
		**<14 Years** **(*n* = 0)**	**14–15 Years** **(*n* = 1)**	**16–17 Years** **(*n* = 11)**	** *p* ** **-Value for Age**	**Male** **(*n* = 6)**	**Female** **(*n* = 6)**	** *p* ** **-Value for Sex**
	**%**	**%**	**%**	**%**		**%**	**%**	
**Urinary cotinine ****					0.500 ^b^			0.286 ^b^
<20 ng/mL	16.7	-	0	18.2		0	33.3	
20–100 ng/mL	33.3	-	100	27.3		50.0	16.7	
>100 ng/mL	50.0	-	0	54.5		50.0	50.0	
**Variables**	**Total** **(*n* = 99)**	**Age Group**				**Sex**		
		**<14 Years** **(*n* = 11)**	**14–15 Years** **(*n* = 29)**	**16–17 Years** **(*n* = 59)**	** *p* ** **-Value for Age**	**Male** **(*n* = 51)**	**Female** **(*n* = 48)**	** *p* ** **-Value for Sex**
	**%**	**%**	**%**	**%**		**%**	**%**	
**GTS in lifetime ***					0.501 ^b^			0.197 ^b^
Never	75.5	72.7	85.7	71.1		82.4	68.0	
1–2 times	11.2	18.2	3.6	13.6		9.8	12.8	
3 times or more	13.3	9.1	10.7	15.3		7.8	19.2	
**Pesticide poisoning in lifetime**					1.000 ^b^			0.243 ^b^
No	97.0	100	96.5	96.6		94.1	100	
Yes	3.0	0	3.5	3.4		5.9	0	
**Coughing without having a cold**					0.048 ^b^			0.679 ^a^
No	83.8	72.7	96.5	79.7		82.4	85.4	
Yes	16.2	27.3	3.5	20.3		17.6	14.6	
**Wheezing in the last 12 months**					0.835 ^b^			0.105 ^b^
No	93.9	90.9	96.5	93.2		98.0	89.6	
Yes	6.1	9.1	3.5	6.8		2.0	10.4	
**Neck pain in the last year**					0.324 ^b^			0.110 ^b^
No	97.0	90.9	100	96.6		100	93.7	
Yes	3.0	9.1	0	3.4		0	6.3	
**Thoracic spine pain in the last year**					0.396 ^b^			0.524 ^a^
No	73.7	90.9	69.0	72.9		76.5	70.8	
Yes	26.3	9.1	31.0	27.1		23.5	29.2	
**Low back pain in the last year**					0.041 ^b^			0.979 ^a^
No	70.7	81.8	86.2	61.0		70.6	70.8	
Yes	29.3	18.2	13.8	39.0		29.4	29.2	

* 1 missing data for this variable. ** 12 observations for this variable. ^a^ Chi-square Test for Heterogeneity ^b^ Fisher’s Exact Test.

## Data Availability

The data presented in this study are not publicly available due to ethical reasons. Data are available from the corresponding author upon reasonable request.

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
