# Peer review of "Child Labor in Family Tobacco Farms in Southern Brazil: Occupational Exposure and Related Health Problems"

_ijerph, 2021, doi:10.3390/ijerph182212255_

Round 1
Reviewer 1 Report
The study entitled “Child Labour in Family Tobacco Farms in Southern Brazil” by Anaclaudia Gastal Fassa et al. investigated the impact of tobacco farming work on the health of children and adolescents by examining the level of urinary cotinine, the occurrence of GTS, pesticide poisoning, respiratory symptoms, and musculoskeletal disorders. This is a very important topic and the current report add valuable information to the existing literature. Overall, this study was well designed with an appropriate statistical analysis on the data. The results were properly interpreted and clearly presented. The reviewer only has one minor concern.
The review has identified a couple limitations of the study as outlined below and hope the author would acknowledge/address them in the discussion.
(1) Urine samples were only obtained from 12 of the 99 young people studied, and no samples from smoker/former smokers. This is a limitation and should be mentioned accordingly.
(2) The authors should also provide a brief literature review on the scientific rationale of measuring urinary cotinine in this study and how it relates to nicotine exposure.
Reviewer 2 Report
This study consists of a social analysis of child labour at Family Tobacco Farms in São Lourenço do Sul (southern Brazil). The amount of information collected by the interviewees and subsequently analysed together with some analytical factors (urine samples) is considerable. However, some issues of the study need to be addressed and improved.
First of all, I have some general comments:
- It is recommended to include as supplementary material the raw data obtained during the interviews, as well as examples from the questionnaires.
- The tables provide valid and necessary information for the understanding of the article. However, some explanatory figures could be considered to facilitate the reading of the results.
- The study carried out to determine "pesticide poisoning", although really interesting, should be reinforced with analytical data obtained from biomonitoring (with urine samples again, or by analysis of passive samplers such as patches, for example). I suggest that this limitation in the study should at least be mentioned in the discussion of the results.
- Regarding the sampling and data collection date for the study (2011): in line 324-326 you justify the absence of changes in laws and regulations as a pretext to assume that there are no considerable changes in the results obtained. However, has this been corroborated in an appropriate way to provide a valid justification for this assumption? Were all possible factors that could influence the results of the study over the last 10 years taken into account? (social context, major events such as the COVID pandemic, etc.)
Specific comments:
A title that provides more information about the study is recommended. My proposal: "Child labour on family tobacco farms in southern Brazil: occupational exposure and related health problems"
Lines 65-68 describe the overall objective/methodology of the study and I consider that they should be included in the Abstract Section.
Materials and Methods
- Including an example of the questionnaire in supplementary material would be helpful for the reader to understand all aspects taken into account in the interviews.
- Important information on the analytical methodology used on urine samples (to detect cotinine) is missing. More detailed description of the sample analysis technique (extraction and subsequent detection and quantification) is required. The prior preservation of the sample should also be included, as well as consideration of possible matrix effects and the QA/QC methods used. This information could be included as supplementary material.
Results
- In general, the presentation of the results is disorganised. The characteristics of the interviewees should be stated first, followed by the working conditions, functions performed, pollutants/chemicals with which they are in contact, etc. The percentages should be reduced (refer to tables).
- Line 136: " 92.6% of the farmers...". Do not start sentences with numbers. Write, for example, "about 93% of the farmers...".
- Line 143: How exactly are passive smokers considered?
- Line 191-193: The absence of personal protective clothing should be indicated along with the other considerations of working conditions.
- It is recommended to write % from highest to lowest (reading coherence). Example in lines 195 - 201..
Discussion
- Lines 234 - 236: Specify where laws apply. Regulations used.
- To include more studies related to pesticide exposure. In addition, highlight the limitations assumed by your study (pesticide/metabolite levels in workers are not analysed).
Round 2
Reviewer 2 Report
Thank you for your response to the comments made during the review process.
Most of the suggestions and proposed corrections have been addressed well.
However, I see the need to add as supplementary material an outline/figure/table summarising the questions asked by the interviewers. In addition, Figure S1 could be added to the main body of the manuscript.
I believe that with these corrections, the article would be ready for acceptance.
Kind regards.
